# Research on the Safety and Security Distance of Above-Ground Liquefied Gas Storage Tanks and Dispensers

**DOI:** 10.3390/ijerph19020839

**Published:** 2022-01-12

**Authors:** Bożena Kukfisz, Aneta Kuczyńska, Robert Piec, Barbara Szykuła-Piec

**Affiliations:** 1Faculty of Security Engineering and Civil Protection, The Main School of Fire Service, 52/54 Słowackiego Street, 01-629 Warsaw, Poland; bpiec@sgsp.edu.pl; 2Institute of Safety Engineering, The Main School of Fire Service, 52/54 Słowackiego Street, 01-629 Warsaw, Poland; akuczynska@sgsp.edu.pl; 3Institute of Internal Security, The Main School of Fire Service, 52/54 Słowackiego Street, 01-629 Warsaw, Poland; rpiec@sgsp.edu.pl

**Keywords:** flammable liquid storage tank, safety and security distance, disaster prevention

## Abstract

Many countries lack clear legal requirements on the distance between buildings and petrol station facilities. The regulations in force directly determine the petrol station facilities’ required distance to buildings, and such distances are considered relevant for newly designed and reconstructed buildings. Public buildings must be located no closer than 60 m to the above-ground liquefied gas tanks and liquid gas dispensers. Still, based on engineering calculations and the applied technical measures, it is possible to determine a safe distance for buildings that are constructed, extended and reconstructed, to which superstructures are added or whose utilisation method changes. The paper presents the results of calculations devoted to determining a safe distance between public buildings and LPG filling station facilities, using selected analytical models. The analyses were carried out for the LPG gas system commonly used in petrol stations, consisting of two gas storage tanks of 4.85 m^3^ capacity each, and a dispenser. It is legitimate to eliminate the obligation to observe the 60 m distance between LPG filling stations and public buildings and the mandatory distance of 60 m between liquefied gas dispensers and public buildings is not justified in light of the implemented requirements to use various protections at self-service liquefied gas filling stands.

## 1. Introduction

Cars powered with gas have been known since the beginning of the automotive industry when, in 1860, Etienne Lenoir constructed the first combustion engine using a mixture of coal gas (used for illuminating streetlamps) and air. Liquefied gas, being a mixture of two LPGs, was first obtained in 1910 by an American chemist, Dr Walter O. Snelling, and was used for powering a car three years later [1]. LPG has maintained its popularity as fuel powering car engines due to its economic aspect—the LPG to petrol price ratio is still favourable, despite the tax burdens introduced for LPG. Lower emissions to the environment, including but not limited to carbon dioxide and nitrogen oxides, are another vital aspect. LPG is then regarded not only as cheap but also as an environmentally friendly fuel.

The name LPG (liquefied petroleum gas) applies to a mixture of liquefied hydrocarbon gases, mainly propane and butane, which also contains small quantities of propene, butene and C_5_ hydrocarbons. Propane–butane is primarily obtained from crude oil cracking, namely processing its heavy fractions into petrol, oils and crude oil hydrogenation. Smaller quantities of LPG are obtained directly from crude oil and natural gas wells, typically when a new well begins to operate. LPG is then a natural side product of crude oil production.

Poland is ranked as the leading LPG consumer in the European Union and the fourth country in the world with the highest LPG consumption for transport purposes, alongside South Korea, Turkey and Russia, which hold the top positions [2]. The LPG market is saturated, and the state and international organisations’ policy that supports electric vehicles leads to reduced interest in LPG systems.

An increased number of adverse events at LPG filling stations has been observed recently. This can be attributed to a higher number of LPG filling modules at petrol stations and LPG filling self-service. According to the statistics of events published on the Internet between 2005 and 2020, leaking systems or failure of the system components were the leading causes of dangerous incidents at LPG filling stations. The statistics revealed that valves and gaskets were the components of the LPG storage tanks’ systems most susceptible to failures. In 90% of the cases, the system components’ leakage or failure resulted in gas emission to the atmosphere. The second most common cause of failures is a vehicle running over an LPG system component (14% of incidents) and pulling the hose out of a dispenser. Both seem to be caused by unintentional errors of drivers filling their vehicles. Dispensers feature embedded protection to cut off the gas supply instantly and prevent such occurrences. Every tenth failure results from leakage (leakage of the tank truck, leakage of the tank truck hose, storage tank filling valve defect or hose leakage while filling the vehicle) [3,4].

Unfortunately, an LPG system’s leaking component, e.g., a hose, does not provide information on the fundamental cause of the leakage (such as manufacturing defect, mechanical damage, or improper maintenance). Service errors cause failures at LPG filling stations less often. They tend, however, to result in more severe failures—jet fires can occur, e.g., after valve replacement. Boiling liquid expanding vapour explosions (BLEVE) and fireballs are observed in exceptional situations. Other causes of failures are observed incidentally (intentional action and hydrostatic valve activation—ca. 1.2% of events).

A typical LPG filling system (Figure 1) consists of the following components:✓Two gas storage tanks, 4.85 m^3^ volume each,✓An electric motor-powered pump, Ex design, with steel sheet housing,✓Liquefied gas dispenser,✓Piping with adequate valves.

Due to its physical and chemical characteristics, LPG poses a severe hazard for humans and the surrounding infrastructure. Many countries have not developed uniform standards for gas filling station construction, which causes many problems such as inappropriate location, inadequate management, potential risk and lack of supply-demand balance [5]. The determination of a safe distance to liquid fuel filling station facilities requires knowledge of the potential causes of failures involving LPG, as well as understanding the complexity of the phenomena that occur during such failures and their effects [6]. The initial occurrence which triggers a series of adverse events is always the cause of the failure. Human errors might include such causes as tearing the vehicle filling hose, gas leakage during gas pumping from the tank truck to the storage tank due to incorrectly connected or broken hose of the tank truck, damage to above-ground tank systems and the tanks by a vehicle running into them, gas dispenser overturning and damage, and incorrect technical service, e.g., during maintenance works. The system failures are also classified as initial occurrences and include leaking flanged connections, valve malfunction and material defects. Adequate gas concentration in the mixture with air is the prerequisite for fire or explosion. If the concentration is close to the lower explosive limit (LEL) of vapours, which amounts to ca. 1.9% volume for LPG, fire is likely to occur when the vapour cloud is ignited. An explosion is possible if the concentration of the flammable substance dispersed in the air falls within the explosive concentration range, i.e., between the lower and upper explosive limit, that is between 1.9 and 9.6% vol. A source of ignition is indispensable for fire or explosion to occur; the sources of ignition at a petrol station include the hot surface of a vehicle engine, short-circuit sparks generated due to failure of the engine’s mechanical components, static electricity, seizure of the pump motor bearings, atmospheric discharge, open flame, stray currents and cathode-based corrosion protection.

There are many scenarios of failure development at a petrol station, depending on the release conditions, local weather conditions (wind direction and speed, temperature, time of day) and topography (surface roughness, natural topography and land development) [7]. The possible failure scenarios at LPG filling stations include jet fire (JF), vapour cloud dispersion (VCD), flash fire (FF), vapour cloud explosion (VCE), unconfined vapour cloud explosion (UVCE), pool fire (PF), BLEVE and fireball (FB) [8].

## 2. Materials and Methods

### 2.1. Criteria Applied to Determine the Safe Distance

With regard to the lack of guidelines on the limit values of potential failure effects at LPG filling stations, literature data and probit functions were used in the study to determine their values. Since a propane-butane gas mixture is not classified as toxic, the effects of LPG storage tank failures were analysed for the influence of overpressure wave (UVCE, BLEVE) and thermal radiation flux (jet fire, fireball caused by BLEVE). The adopted thermal radiation and overpressure limits helped determine the safe distance of public buildings to petrol stations.

The duration of exposure to radiation matters in the thermal radiation analysis. For persons staying within the hazard impact zone, the duration is typically 10 s (as the person is assumed to find a place to hide within such time) or 30 s for persons who do not escape immediately or if no PPE is provided [9]. It seems justified to adopt such duration for public buildings as they do not include buildings intended for people with limited mobility, e.g., hospitals, creches, kindergartens or elderly care centres. Most people staying in the building are assumed to be able to find a hiding place in case of hazard quickly. The thermal radiation impact for a fireball is much shorter. According to the calculations carried out in areal location of hazardous atmospheres (ALOHA), when an 85% filled LPG storage tank explodes, a fireball lasts for 6 s. The duration of 10 s and 30 s applies to an analysis of thermal radiation caused by jet fires, being pool fires.

Probit functions are among the statistical models used for determining the likelihood of occurrence of a specific injury type suffered by people, e.g., as a result of exposure to thermal radiation. The probit functions are determined based on an analysis of injuries resulting from specific occurrences or derived from research. They determine the percentage of people who suffer from a specific type of injury following exposure to a particular type of load [9,10,11].

The general form of a probit function is as follows:*P*_*r*_=*A* + *B ln (L)*(1)
where:

*A*—constant depending on the type of injury and type of load,

*B*—constant depending on the type of load,

*L*—load (dose).

The probit functions shown in Table 1 refer to an unprotected body and will be used for short-term heat exposure. Moreover, casualties are included in the lower classes (Grade I and Grade II burns). The same applies to people suffering from Grade II burns—they are also included in the group with Grade I burns. This means that the specific probit function determines the likelihood of a particular population percentage suffering from at least that particular injury.

The functions above were used for determining the part of the population that would suffer from a particular type of injury as a result of exposure to the specific thermal radiation value. The durations of 10 s and 30 s were adopted for the calculation, corresponding to the jet fire or pool fire impact duration (Table 2), while the durations of 5 s and 6 s for fireball impact (Table 3) resulting from BLEVE, respectively for 42.5% and 85% of the storage tank volume filled.

The value of 4.7 kW/m^2^ was taken as the limit parameter of the heat impact caused by jet fire; reaching the value marks the safe distance to public buildings. For the quoted thermal radiation value, most of the instantly evacuated people will experience no pain. Other people may experience pain; however, assuming that everybody finds a place to hide within 30 s, no severe burns are expected. During 30 s of exposure to a thermal radiation value of 4.7 kW/m^2^, about 30% of the population may suffer from Grade I burns [12].

The thermal radiation value of 12.6 kW/m^2^ with a shorter duration was adopted for a fireball as the criterion to determine the safe distance. At such thermal radiation flux density value and short exposure time, 10% of the population may suffer from Grade I burns [13].

The limit value adopted for overpressure amounts to 3.5 kPa. Such an overpressure value does not pose a direct hazard for the human body. It may, however, cause cracking of glass panes and minor damage to some parts of buildings, which in turn may lead to human injuries (by flying glass splinters) [14].

If flammable gas is released into the atmosphere and a flammable gas cloud is formed due to non-homogeneous mixing of the gas with air, only a small part of the flammable gas might fall within the concentration range between the lower and upper explosive limits. That is why only part of the cloud may ignite and a flash fire occur. ALOHA does not model flash fires, but it estimates the highly flammable area of the vapour cloud—this is the area where flash fire may occur some time after the release has started. The LEL can be expected to be used in order to determine the areas where flash fire may occur. The concentration levels estimated by ALOHA are time averaged. The real cloud includes the areas where concentration exceeds the average and areas where concentration is lower than the average. Due to non-uniform concentration (patchiness), some areas called pockets occur, where the chemical substance stays in the highly flammable area, even though the mean concentration drops below the upper explosive limit (UEL). Some experiments revealed that flame pockets might occur in spaces where the mean concentration exceeds 60% of the UEL, and that was the value taken as the limit [15].

### 2.2. Hazard Analysis

When analysing the risk related to the potential fire or explosion occurrence at an LPG filling station, one must remember that the propane–butane gas mixture has been qualified as hazardous material in the explosion Group IIa. Explosion hazard zones are identified for filling station facilities. Explosion hazard zone formation and the zones’ dimensions are summarised in Table 4.

As shown in Table 4, the explosive atmosphere’s radius near the above-ground LPG storage tank is up to 1.5 m from all connector pipes of the storage tank in normal conditions of use. It should be added that explosion hazard zones do not occur outside the dispenser but only inside and within the safety gap. In the scenarios adopted for this analysis, pressurised liquefied gas is stored in two above-ground tanks of 4.85 m^3^ volume each. The probability of substance ignition in the storage tank is low. LPG storage tanks are typically coated with paints that reflect at least 70% of thermal radiation, which protects them against excessive heat from solar radiation. Safety valves on top of the storage tanks release gas excess to the atmosphere in a controlled way if the pressure inside the tank rises.

#### 2.2.1. Piping Failure or Damage

The uncontrolled release of LPG is typically caused by a system component’s leakage or failure. The piping diameters used for the liquid and gaseous phase include DN15, DN20, DN25 and DN32. Gas escape through a 32 mm diameter hole/pipe was adopted for the calculations. Two cases of storage tank failure or damage were verified:✓Compromised integrity of the storage tank at the DN32 duct—probable in case of service works, which often involve replacement of the pipeline components, e.g., filters or valves,✓Compromised integrity of a storage tank with a DN32 hole—release directly from the hole in the storage tank. This is a less probable but still possible case, e.g., when a vehicle hits the system and pulls out the entire hose.

The calculations were carried out for the maximum acceptable tank filling, amounting to 85%. For calculation purposes, the gas inside the tank was assumed to be propane (due to its physical and chemical characteristics, including lower boiling point and higher volatility), which is more dangerous than butane. The ALOHA program was used for modelling the hazard. The program allows the entering of details concerning the actual or potential release of substances and then generates estimates of the hazard zones in the form of their toxicity, thermal radiation or overpressure limit values, depending on the type of hazard. The parameters used in the calculations were as follows:

Wind speed: 5 m/s,

Ambient/liquid gas temperature in the tank: 20 °C,

Stability class (A–F): D,

Terrain roughness: ≤20 cm,

Cloudy: medium,

Humidity: 80%.

#### 2.2.2. Vehicle Running into a Gas Dispenser

A vehicle hitting a dispenser is the second most common cause of uncontrolled gas escape. Liquid fuel dispensers should be protected against vehicles running into them by being placed on isles raised at 0.15 m or in another effective way. Additional barriers are another popular solution. Emergency release couplings installed at the ground level, at the dispenser base, provide the system’s essential protection if a vehicle runs into the dispenser; once damaged, they block gas release automatically. Hitting a dispenser with a vehicle typically results in the emission of gas that has remained inside the dispenser and filling hose. The statistics confirm that, due to the current protective measures, if such an event occurs, no further hazards arise besides a relatively low gas emission to the atmosphere.

#### 2.2.3. Pulling Out the Filling or Draining Hose

Pulling out the filling hose when a vehicle drives away during filling with the hose still attached, is another cause already mentioned above. This may apply both to vehicles being filled and to tank trucks discharging liquefied gas; nonetheless, no such case has been presented in the statistics for tank trucks. In order to prevent excessive leaking of a dangerous substance, LPG filling and draining hoses feature emergency release couplings. The coupling is disconnected before the hose is damaged, and the valves inside the coupling close automatically, cutting off substance release. Owing to this solution, if a filling hose is pulled out, only a relatively small amount of gas is emitted and, consequently, no fire or hazard occurs. If a dispenser hose (16 mm diameter, 40 m long) or an LPG drain hose (50 mm diameter, 40 m long) is pulled out, ca. 0.73 kg and 47.27 kg of LPG can be released respectively. In order for fire to turn into UVCE, a minimum weight of flammable material is necessary, i.e., the critical mass. Fuel critical masses vary significantly, and it is difficult to determine a single critical mass for each material. Generally, this is estimated as 1000–15,000 kg of fuel [17], which means that it is much more than the gas mass that can escape from the dispenser hose or tank truck hose. Therefore, no UVCE should be expected if the hose is pulled out and the valves close immediately.

#### 2.2.4. BLEVE

For the BLEVE scenario, the explosion was assumed to occur due to the LPG storage tank heating by jet fire to 40 °C. It is worth adding that it is often possible to estimate the BLEVE effects using very simple expressions of the acoustic loudness source [18]. As reflected by more recent studies, more accurate far-field blast pressure prediction could be achieved by conducting computational fluid dynamics (CFD) simulations using the models developed by Van den Berg et al. [18]. However, in these models, it was assumed that the blast is resulted from the instantaneous phase change of liquid and vapour expansion, which is in contradiction with the statement in the publication [19] that the vapour expansion is the main source of shock wave generation while the liquid evaporation is not as fast as the vapour expansion.

All scenarios described in items 2.2.1–2.2.4 are summarised in Table 5.

## 3. Results

Table 6 summarises the thermal radiation range for the analysed scenarios—jet fire and BLEVE.

Table 7 summarises the blast wave ranges for UVCEs. The ALOHA program does not generate the overpressure value for BLEVE.

Table 8 summarises the flammable gas cloud range where flame pockets may occur.

## 4. Discussion

According to an analysis carried out in ALOHA for representative jet fire scenarios (1-JF, 2-JF), the range for which the hazard source generates radiation higher than the assumed limit value of 4.7 kW/m^2^, is from 29 m to 43 m. For BLEVE scenarios (7-BLEVE, 8-BLEVE, 9-BLEVE) and short-term radiation emission, the limit value of 12.6 kW/m^2^ is reached for distances up to 149 m in the case of an 85%-filled LPG storage tank explosion. The calculations revealed that BLEVE and its accompanying fireball are deemed disasters in light of the current regulations. At the distance of 60 m, which according to current effective regulations is regarded as safe from the point of view of the explosion source location from a public building, the forecasted thermal flux value exceeds 35 kW/m^2^. Burns in nearly the entire population are expected for such a thermal radiation value, and ca. 15% of the population may die (Table 3). Draft acts of law concerning the method of determining safe distance to plants that pose a hazard of severe industrial failure within the range of 37.5 kW/m^2^ radiation value, it is possible to locate commune roads and farmland not including any buildings intended for human stay or animal breeding. For radiation values up to 12.5 kW/m^2^, it is possible to locate railway lines, production and storage facilities, other types of public roads (including national roads), and plants that pose a severe industrial failure hazard. Low residential buildings and public buildings with an area up to 500 m^2^ can be located within zones where the expected thermal radiation flux is lower than 12.5 kW/m^2^. This shows an effort to prevent such failures. Should they occur, they are regarded as disastrous events and their highly adverse effects are accepted.

An analysis of flammable gas cloud explosions for representative scenarios revealed that the overpressure limit value of 3.5 kPa occurs within the 42–66 m range from the failure source. This means that for the most unfavourable scenario, the immediate risk to human life is no longer present at a distance greater than 66 m. Airborne glass splinters, however, still pose a risk.

According to the data shown in Table 3, for the adopted scenarios at a distance up to 103 m from the storage tank, areas with flammable gas concentration between the lower and upper explosive limits may occur in the flammable gas cloud; once these areas are ignited, a flash fire may occur. ALOHA cannot indicate these areas, as they vary in time and the resultant values from ALOHA are only averaged. Staying inside the flame pocket when it becomes ignited poses the risk of injuries.

The 60 m distance to public buildings also applies to liquefied gas dispensers. Since 2013, vehicle users have been allowed to fill their vehicles with LPG on their own. This entails the implementation of new requirements to improve the safety of the filling station and filling process, so provisions were added to the legal acts stipulating that liquefied gas dispensers at liquid fuel filling stations and self-service liquefied gas filling stations will feature a filling switch controlling the opening and closing of the cut-off safety valve that prevents liquefied gas release in case of emergency. The filling nozzle design will prevent the filling nozzle’s valve from opening if incorrectly blocked or sealed at the connection with the vehicle filling system. The maximum liquefied gas release to the atmospheric air while uncoupling the filling nozzle from the vehicle will not exceed 1 cm^3^. Stations are equipped with alarm systems that generate an acoustic signal owing to which the user can inform the petrol station’s staff about an emergency situation. The staff, in turn, should be monitoring the station continuously. Automatic gas flow cut-off valves must be installed between the dispensers and the process piping that supply the liquefied gas dispensers at liquid fuel filling stations and self-service liquefied gas filling stations. The cut-off valves should be strictly linked with the operation of the liquefied gas dispensers so that when the LPG dispensers switch on, the cut-off valves should open, and if the LPG dispensers are switched off, the cut-off valves should close.

The abovementioned provisions indicate that self-service LPG filling stations for vehicles must fulfil a number of additional requirements that were not considered when determining the mandatory 60 m distance between a liquefied gas dispenser and public buildings. Moreover, a liquefied gas dispenser at a vehicle filling station should be protected against a vehicle running into it by being placed on an isle raised 0.15 m above the adjacent access ramp or in another effective way. The dispenser should also come with a cut-off valve preventing liquefied gas leakage in case of emergency. The hose connecting the LPG dispenser with the vehicle being filled must also be protected against a vehicle running over it and feature a cut-off valve (emergency disconnect coupling).

## 5. Conclusions

The conclusions must be preceded by the following caveat: the scenarios analysed with the ALOHA program were highly unfavourable. Gas release through 3.2 cm holes was considered in the analysed scenarios because of the sizes of valves and hoses used in LPG filling systems. If the valves or couplings lose their integrity, the hole through which the gas escapes to the environment may be smaller. Nonetheless, according to the conventional methods, analyses should consider the most pessimistic, possible failure scenario. The assumption of a gas leak at the bottom of the storage tank filled to its maximum volume was the evidence of such an approach; the leakage spot in the presented location affects the gas release rate. There is no official register of failures at LPG filling stations, which hampers the development of reliable statistical analyses. Considering the recent growth of the Polish market, reaching one of the highest global values, such negligence is shocking. Moreover, there are no standardised legal acts on determining a safe distance to plants posing a risk of severe industrial failure, which blurs the understanding of what safe distance actually means. The definition of “safe distance” in many valid regulations is not clear and must be specified through engineering calculations. The safe distance between public buildings and petrol stations is eventually determined axiomatically, depending on the acceptable risk level [20,21,22].

Technical and equipment factors and direct and indirect human errors have an influence on risk perception and impaired mobility of certain individuals might play a pivotal role in contributing to safety and security at LPG stations. The oil and gas industry has been beset with several catastrophic accidents, most of which have been attributed to organisational and operational human factor errors. The human factors analysis and classification system for the oil and gas industry (HFACS-OGI) is effective for the analysis of human factors, particularly as it relates to safety culture, management commitment, safety leadership, organisational erosive drift, technical failure of ageing equipment and the operators’ lack of knowledge or competency, failures in national and international industry regulatory standards and emerging violation issues such as sabotage, in response to problematic organisational factors particular to the oil and gas industry. These industry standards and national regulations could have provided the reference guidelines for communication, delegation of authority, human resources policies, positive norms, organisational customs, values and beliefs to enable them to operate safely, but deficiencies of industry codes and national regulations have been identified as key factors that may both prevent and also contribute to accidents [23].

Considering the assumption that many people with impaired mobility are not expected to be present in public buildings, it is justified to determine the hazard zone at the border of which the heat source generates radiation up to 4.7 kW/m^2^. For the purpose of determining safe distance, it is justified to adopt the thermal radiation limit value of 12.6 kW/m^2^ for a very short exposure to thermal radiation. The presented conclusions reveal the need to instigate legislative works to regulate the legal status, especially in order to eliminate the obligation to observe the 60 m distance between LPG filling stations and public buildings and to reduce the value to e.g., 45 m, which is the distance that protects against typical and most probable hazards at LPG filling stations. The mandatory distance of 60 m between liquefied gas dispensers and public buildings is not justified in light of the implemented requirements to use various protections at self-service liquefied gas filling stands.

The analysis also highlights that legal regulations do not accept the reduction of the distance between above-ground LPG storage tanks and public buildings if a fire partition is used. Making reference to the rules resulting from petrol and LPG filling stations’ operation, including fire statistics, the admissible reduction of the mandatory distance by half, from 30 to 15 m, is substantiated if a fire partition is used. Rejecting such a possibility in the current legal status seems surprising.

## Figures and Tables

**Figure 1 ijerph-19-00839-f001:**
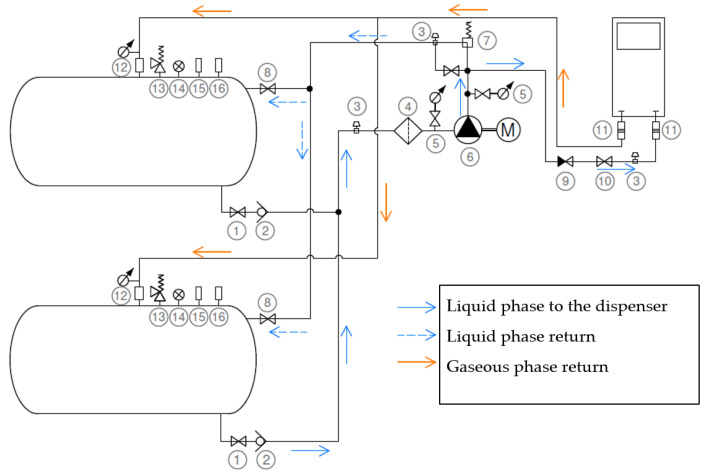
Flow diagram of liquid gas filling system, where (1) liquid phase intake ball valve, (2) overflow valve, (3) hydrostatic valve, (4) liquefied gas filter, (5) dial pressure gauge, (6) liquefied gas pump, (7) by-pass overflow valve, (8) liquid phase intake valve, (9) non-return valve, (10) ball valve, (11) emergency release coupling, (12) gaseous phase intake valve with pressure gauge, (13) safety valve, (14) tank filling indicator, (15) valve of tank filling by tank truck, (16) storage tank emergency drain valve.

**Table 1 ijerph-19-00839-t001:** Forms of probit functions for heat flux impact effects [9], where: *t*—exposure duration [s]; *q*—heat load [W/m^2^].

Parameter	Constant Depending on the Injury Type and Load Type—A	Constant Depending on the Load Type—B	Load (Dose)—L	Probit Function form for Constant Thermal Radiation
Grade I burn	−39.83	3.0186	*tq^4/3^*	$P_{f}=-39.83+3.0186ln\;\left(tq^{4/3}\right)$
Grade II burn	−43.14	3.0188	*tq^4/3^*	$P_{f}=-43.14+3.0188\text{ln}\;\left(tq^{4/3}\right)$
Death	−36.38	2.56	*tq^4/3^*	$P_{f}=-36.38+2.56\text{ln}\;\left(tq^{4/3}\right)$

**Table 2 ijerph-19-00839-t002:** Likelihood of occurrence of a specific type of injury suffered by persons due to exposure to thermal radiation for 10 s and 30 s caused by jet fire or pool fire.

Thermal Radiation Flux Density [kW/m^2^]	Grade I Burn after 10 s	Grade I Burn after 30 s	Grade II Burn after 10 s	Grade II Burn after 30 s	Death after 10 s	Death after 30 s
4.7	0	30	0	0	0	0
7.0	2	86	0	2	0	1
9.5	16	99	0	16	0	8
12.6	55	100	0	55	0	33
23.0	99.5	100	22	99.5	12	95
35	100	100	82	100	59	99.9

**Table 3 ijerph-19-00839-t003:** Likelihood of occurrence of a specific type of injury suffered by persons due to exposure to thermal radiation for 5 s and 6 s caused by fireball.

Thermal Radiation Flux Density [kW/m^2^]	Grade I Burn after 10 s	Grade I Burn after 30 s	Grade II Burn after 10 s	Grade II Burn after 30 s	Death after 10 s	Death after 30 s
4.7	0	0	0	0	0	0
7.0	0	0	0	0	0	0
9.5	0	1	0	0	0	0
12.6	3	8	0	0	0	0
23.0	68	85	1	2	0	0
35	99	99.7	13	27	6	14

**Table 4 ijerph-19-00839-t004:** Explosion hazard zone for the normal operation of a filling station according to Annex 1 of Regulation [16].

Name of the Explosion Hazard Area	Explosion Hazard Category	Dimensions of the Explosion Hazard Zones, Calculated from the Hazard Sources
Above-ground storage tank, up to 10 m^3^ volume	2	Within 1.5 m from all storage tank connection pipes
Reloading station from the tank truck	2	Within 1.5 m from the tank truck discharge coupling
LPG dispenser	1	Inside the dispenser’s hydraulic part and in the pit underneath the dispenser
LPG dispenser	2	Inside the safety gap

**Table 5 ijerph-19-00839-t005:** Failure scenarios.

Scenario Designation	Scenario Description
1-JF	Jet fire as a result of compromised integrity of a storage tank with DN32 hole
2-JF	Jet fire as a result of damage to DN32 diameter hose
3-FF	Formation of a flammable gas cloud and its potential fire due to compromised integrity of a storage tank with DN32 hole
4-FF	Formation of a flammable gas cloud and its potential fire due to damage to DN32 diameter hose
5-UVCE	Flammable gas cloud explosion as a result of compromised integrity of a storage tank with DN32 hole
6-UVCE	Flammable gas cloud explosion as a result of damage to DN32 diameter hose
7-BLEVE	BLEVE fire if 500 kg of LPG is left in the storage tank—ca. 20% of the storage tank volume
8-BLEVE	BLEVE fire if 1000 kg LPG is left in the storage tank—ca. 42.5% of the storage tank volume
9-BLEVE	BLEVE fire if the LPG storage tank is filled to its maximum level (85% of the storage tank volume)—implausible scenario. Such an occurrence could be possible if a vehicle hit the storage tank

**Table 6 ijerph-19-00839-t006:** Thermal radiation range.

Thermal Radiation Flux Density (kW/m^2^)		35.0	23.0	12.6	9.5	7.0	4.7	2.1
Scenario Designation	Time (s)	Thermal Radiation Range (m)						
1-JF	180	13	19	26	31	36	43	63
2-JF	420	10	12	18	20	24	29	42
7-BLEVE	4	55	71	97	112	131	159	235
8-BLEVE	5	70	89	123	141	165	201	297
9-BLEVE	6	85	109	149	173	201	245	362

**Table 7 ijerph-19-00839-t007:** Blast wave range.

Overpressure [kPa]		7.0	6.0	5.0	4.0	3.5
Scenario Designation	Time [s]	Blast Wave Range [m]				
5-UVCE	180	Not reached	63	64	65	66
6-UVCE	420	Not reached	39	40	41	42

**Table 8 ijerph-19-00839-t008:** Flammable gas cloud range (specified concentration).

Concentration Value [% vol.]	LEL	60% LEL	10%LEL
Scenario Designation	Flammable Gas Cloud Range [m]		
3-FF	76	103	295
4-FF	47	63	187

## Data Availability

This statement if the study did not report any data.

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
