# Peer review of "Research on the Safety and Security Distance of Above-Ground Liquefied Gas Storage Tanks and Dispensers"

_ijerph, 2022, doi:10.3390/ijerph19020839_

Round 1

Reviewer 1 Report

A brief sentence or two highlighting the findings of this study should be included in the abstract.

Lines 50-82: Can there be adequate references to back up these statistical claims made in this paragraph? Lines 52-53 read “According to the statistics of events published on the Internet between 2005 and 2020”; however, can the exact source of this data be highlighted?

In line 68, the full meaning of BLEVE should be stated when first mentioned (with BLEVE in a bracket), after which the abbreviation can be used subsequently in the paper. This was done in line 113-114 but should have been done earlier.

In line 132, the full meaning of ALOHA should be stated when first mentioned (with ALOHA in a bracket), after which the abbreviation can be used subsequently in the paper.

In line 190, the full meaning of UEL should be stated just as it was done for Lower Exposure Limit (LEL) in line 99.

In lines 345-349, your conclusion section highlights the level to which risk perception and impaired mobility of certain individuals might play a pivotal role in contributing to safety and security at LPG stations. While the study analysis has taken cognisance of technical and equipment factors that need to be considered to ensure safety and security of LPG storage tanks and dispensers, it may be worthwhile to ascertain how human errors can be curbed as well. Just as the authors mentioned in lines 54-63, human error is an obvious contributory factor to adverse events at LPG filling stations. Hence, an insight into an analysis from a human factor perspective might be beneficial. It might be worth looking at the Human Factor Analysis and Classification System for the Oil and Gas Industry (HFACS-OGI) framework to provide guidance on certain performance influencing factors that might impede safety and security of LPG gas storage tanks and dispensers.

Author Response

Open Review 1

Comments

I improved and rewrote the manuscript.

  1. A brief sentence or two highlighting the findings of this study should be included in the abstract.

New information had been added to the abstract: It is legitimate to eliminate the obligation to observe the 60 m distance between LPG filling stations and public buildings and the mandatory distance of 60 m between liquefied gas dispensers and public buildings is not justified in light of the implemented requirements to use various protections at self-service liquefied gas filling stands.

  1. Lines 50-82: Can there be adequate references to back up these statistical claims made in this paragraph? Lines 52-53 read “According to the statistics of events published on the Internet between 2005 and 2020”; however, can the exact source of this data be highlighted?

Two literature sources had been added to the bibliography:

  • Uliasz The situation of buildings in the vicinity of LPG stations - determining the range of threats and nuisance as well as methods of their estimation and reduction, Training conference Elements of land development - conditions of fire protection, 28.05.2019, Wroclaw, Poland.
  • Uliasz The situation of buildings in the vicinity of LPG stations - determining the range of threats and nuisance as well as methods of their estimation and reduction, 21st workshop "Fire Safety of Buildings”, 2018, Zieleniec, Poland.

  1. In line 68, the full meaning of BLEVE should be stated when first mentioned (with BLEVE in a bracket), after which the abbreviation can be used subsequently in the paper. This was done in line 113-114 but should have been done earlier.

This part had been changed.

  1. In line 132, the full meaning of ALOHA should be stated when first mentioned (with ALOHA in a bracket), after which the abbreviation can be used subsequently in the paper.

This part had been changed.

  1. In line 190, the full meaning of UEL should be stated just as it was done for Lower Exposure Limit (LEL) in line 99.

This part had been changed.

  1. In lines 345-349, your conclusion section highlights the level to which risk perception and impaired mobility of certain individuals might play a pivotal role in contributing to safety and security at LPG stations. While the study analysis has taken cognisance of technical and equipment factors that need to be considered to ensure safety and security of LPG storage tanks and dispensers, it may be worthwhile to ascertain how human errors can be curbed as well. Just as the authors mentioned in lines 54-63, human error is an obvious contributory factor to adverse events at LPG filling stations. Hence, an insight into an analysis from a human factor perspective might be beneficial. It might be worth looking at the Human Factor Analysis and Classification System for the Oil and Gas Industry (HFACS-OGI) framework to provide guidance on certain performance influencing factors that might impede safety and security of LPG gas storage tanks and dispensers.

New information had been added to the conclusions: Technical and equipment factors  and direct and indirect human errors have an influence on risk perception and impaired mobility of certain individuals might play a pivotal role in contributing to safety and security at LPG stations. The oil and gas industry has been beset with several catastrophic accidents, most of which have been attributed to organisational and operational human factor errors. The Human Factors Analysis and Classification System for the Oil and Gas Industry (HFACS-OGI) is effective for the analysis of human factors, particularly as it relates to safety culture, management commitment, safety leadership, organisational erosive drift, technical failure of ageing equipment and the operators’ lack of knowledge or competency, failures in national and international industry regulatory standards and emerging violation issues like sabotage, in response to problematic organisational factors particular to the oil and gas industry. These industry standards and national regulations could have provided the reference guidelines for communication, delegation of authority, human resources policies, positive norms, organisational customs, values and beliefs to enable them to operate safely, but deficiencies of industry codes and national regulations have been identified as key factors that may both prevent and also contribute to accidents.

Reviewer 2 Report

The paper describes and evaluates the possible hazards of LPG leaks at an LPG filling station for automobiles.

The Introduction is an excellent description of the state of affairs of LPG use, but it is deficit of LPG tank station safety studies done elsewhere, e.g., in the Netherlands by RIVM.

In Chap 2, section 2.1 known probit relations for fire hazards are summarized. For a more comprehensive review of these probit relations it is recommended to refer to "Joaquim Casal, 2017. Evaluation of the Effects and Consequences of Major Accidents in Industrial Plants, 2nd ed. ISBN: 978-0-444-63883-0". All these relations boil down to statistical observations of the nuclear effects, a.o., in Japan.

ALOHA is producing rather conservative results due to the fact that the program has been designed for emergency response commanders who have to judge an event situation and to decide about evacuation of people. Please produce e reference of the ALOHA code and the input conditions you used. 

The acronym UVCE for unconfined vapor cloud explosion was in use in the 1980s but was later simplified to VCE as partial confinement playes an important role.

See for BLEVE blast: A.C. van den Berg, M.M. van der Voort, J. Weerheijm, and N.H.A. Versloot, 2006. BLEVE Blast by Expansion-Controlled Evaporation. Process Safety Progress (Vol.25, No.1) 44-51.

Why didn't you include the risk of a leak of a tank truck resupplying the tank station as the amounts of LPG leaked will be larger, while also a BLEVE due to a fire below the tank on the truck may occur.

In your Conclusions you did not mention a final resulting safe distance value, but you mentioned protection measures without further specifying those. The effect of such measures was not discussed in sections 3 or 4. Why not?

So, the paper ends in some vagueness, which may confuse a reader.

Author Response

Open Review 2

Comments

I improved and rewrote the manuscript.

  1. The Introduction is an excellent description of the state of affairs of LPG use, but it is deficit of LPG tank station safety studies done elsewhere, e.g., in the Netherlands by RIVM.

The introduction presents the statistics of events that took place in the years 2005-2020 at petrol stations in Poland. This statement uses data collected by Capt. Eng. Radosław Uliasz from the Provincial Headquarters of the State Fire Service in Wrocław. The statistics of the events were prepared on the basis of press reports published on the Internet. In addition, all events ended in a fire or explosion and all events from the province. Lower Silesia, regardless of their effects (gas emission / fire / explosion), were obtained from the reports of individual commands that participated in these events.

  1. In Chap 2, section 2.1 known probit relations for fire hazards are summarized. For a more comprehensive review of these probit relations it is recommended to refer to "Joaquim Casal, 2017. Evaluation of the Effects and Consequences of Major Accidents in Industrial Plants, 2nd ed. ISBN: 978-0-444-63883-0". All these relations boil down to statistical observations of the nuclear effects, a.o., in Japan.

This literature item was unavailable to me.

  1. ALOHA is producing rather conservative results due to the fact that the program has been designed for emergency response commanders who have to judge an event situation and to decide about evacuation of people. Please produce e reference of the ALOHA code and the input conditions you used. 

The parameters used in the calculations were as follows:

Wind speed: 5 m/s,

Ambient / liquid gas temperature in the tank: 20°C,

Stability class (A-F): D,

Terrain roughness: ≤ 20 cm,

Cloudy: medium,

Humidity: 80%.

  1. The acronym UVCE for unconfined vapor cloud explosion was in use in the 1980s but was later simplified to VCE as partial confinement playes an important role.

The difference between the explosion of a flammable gas cloud in a confined space (VCE) and an unconfined space (UVCE) is that in the first case, an explosion occurs in a dense space, e.g. in the space of full suspended ceilings with a large number of installations, or inside industrial equipment. Overpressure during VCE can reach 10 atm. and under special conditions of deflagration, it may turn into a much more dangerous detonation (the so-called DDT phenomenon). In the case of UVCE, the pressure increase is slower and its value is approx. 1-1.5 atm.

  1. See for BLEVE blast: A.C. van den Berg, M.M. van der Voort, J. Weerheijm, and N.H.A. Versloot, 2006. BLEVE Blast by Expansion-Controlled Evaporation. Process Safety Progress (Vol.25, No.1) 44-51.

New information had been added.

  1. Why didn't you include the risk of a leak of a tank truck resupplying the tank station as the amounts of LPG leaked will be larger, while also a BLEVE due to a fire below the tank on the truck may occur.

 There are the following methods for determining the extent of the environmental impact of a major accident for the purposes of land use planning:

  1. a method of assessing the consequences of a potential major accident - representative scenarios are adopted for analysis, and in one of the variants, the worst possible accident course - a complex method, not very practical,
  2. a method of assessing the risk of a major accident with specific consequences - based on statistical data, complex, time-consuming and costly; depending on the adopted output data and the method of their processing, it leads to a large discrepancy in results,
  3. the method of typical "safe distances" - based on experts' opinions, data from the occurrences and the experience of users of particular types of installations. It is a tabular method and the adopted safe distances depend on the type of industry or the amount and type of hazardous substances present. This method was found to be the most optimal for the purposes of spatial development planning in order to quickly assess the scale of threats.

Returning to the statistics in the years 2005-2020 at petrol stations in Poland, there were a lot of breakdowns at LPG stations, but all ended only with gas emissions.

In Chap. 2, section 2.2.3. the publication was written: “Pulling out the filling hose when a vehicle drives away during filling with the hose still attached, is another cause already mentioned above. This may apply both to vehicles being filled and to tank trucks discharging liquefied gas; nonetheless, no such case has been presented in the statistics for tank trucks. “

  1. In your Conclusions you did not mention a final resulting safe distance value, but you mentioned protection measures without further specifying those. The effect of such measures was not discussed in sections 3 or 4. Why not?

So, the paper ends in some vagueness, which may confuse a reader.

In Chap. 5. the publication was written: “The presented conclusions reveal the need to instigate legislative works to regulate the legal status, especially in order to eliminate the obligation to observe the 60 m distance between LPG filling stations and public buildings and to reduce the value to e.g. 45 m, which is the distance that protects against typical and most probable hazards at LPG filling stations. The mandatory distance of 60 m between liquefied gas dispensers and public buildings is not justified in light of the implemented requirements to use various protections at self-service liquefied gas filling stands.

The analysis also highlights that legal regulations do not accept the reduction of the distance between above-ground LPG storage tanks and public buildings if a fire partition is used. Making reference to the rules resulting from petrol and LPG filling stations' operation, including fire statistics, the admissible reduction of the mandatory distance by half, from 30 to 15 m, is substantiated if a fire partition is used. Rejecting such a possibility in the current legal status seems surprising.”
